# High-Performance Ammonia QCM Sensor Based on SnO_2_ Quantum Dots/Ti_3_C_2_T_x_ MXene Composites at Room Temperature

**DOI:** 10.3390/nano14221835

**Published:** 2024-11-16

**Authors:** Chong Li, Ran Tao, Jinqiao Hou, Huanming Wang, Chen Fu, Jingting Luo

**Affiliations:** 1School of Electronic Engineering, Huainan Normal University, Huainan 232038, China; 2170218809@email.szu.edu.cn; 2Shenzhen Key Laboratory of Advanced Thin Films and Applications, College of Physics and Optoelectronic Engineering, Shenzhen University, Shenzhen 518060, China; 2310452048@email.szu.edu.cn (J.H.); whm13484263919@163.com (H.W.); chenfu@szu.edu.cn (C.F.)

**Keywords:** quartz crystal microbalance, MXene, Ti_3_C_2_T_x_, NH_3_, SnO_2_, colloidal quantum dots

## Abstract

Ammonia (NH_3_) gas is prevalent in industrial production as a health hazardous gas. Consequently, it is essential to develop a straightforward, reliable, and stable NH_3_ sensor capable of operating at room temperature. This paper presents an innovative approach to modifying SnO_2_ colloidal quantum dots (CQDs) on the surface of Ti_3_C_2_T_x_ MXene to form a heterojunction, which introduces a significant number of adsorption sites and enhances the response of the sensor. Zero-dimensional (0D) SnO_2_ quantum dots and two-dimensional (2D) Ti_3_C_2_T_x_ MXene were prepared by solvothermal and in situ etching methods, respectively. The impact of the mass ratio between two materials on the performance was assessed. The sensor based on 12 wt% Ti_3_C_2_T_x_ MXene/SnO_2_ composites demonstrates excellent performance in terms of sensitivity and response/recovery speed. Upon exposure to 50 ppm NH_3_, the frequency shift in the sensor is −1140 Hz, which is 5.6 times larger than that of pure Ti_3_C_2_T_x_ MXene and 2.8 times higher than that of SnO_2_ CQDs. The response/recovery time of the sensor for 10 ppm NH_3_ was 36/54 s, respectively. The sensor exhibited a theoretical detection limit of 73 ppb and good repeatability. Furthermore, a stable sensing performance can be maintained after 30 days. The enhanced sensor performance can be attributed to the abundant active sites provided by the accumulation/depletion layer in the Ti_3_C_2_T_x_/SnO_2_ heterojunction, which facilitates the adsorption of oxygen molecules. This work promotes the gas sensing application of MXenes and provides a way to improve gas sensing performance.

## 1. Introduction

Sensitive gas detection plays a crucial role in various fields, such as industrial production, environmental monitoring, industrial safety, non-invasive disease diagnostics, and more [1,2]. Among the different gasses, ammonia (NH_3_) is a colorless gas characterized by a pungent and irritating odor, and it is widely used in various industries, including the chemical industry, nitrogen fertilizers, and pharmaceutical industries [3,4]. However, long-term or high-concentration exposure to ammonia can irritate the respiratory system, skin, and eyes and can pose a significant threat to human health, especially in confined spaces [3,5]. The safety limits values for continuous exposure to 25 ppm NH_3_ and 35 ppm NH_3_ are 8 h and 15 min, respectively, as reported of the U.S. Occupational Safety and Health Administration (OSHA) [6]. Therefore, it is particularly important to develop sub-ppm NH_3_ sensors to adapt to the Internet of Things.

Currently, there are various types of ammonia sensors, such as electrochemical gas sensors, optical fiber gas sensors [7], resistor gas sensors [8], surface acoustic wave (SAW) gas sensors [9], and quartz crystal microbalance (QCM) gas sensors [10]. Among them, QCM is a kind of highly sensitive mass sensor that can calculate the adsorbed weight of the sensor film by detecting the working frequency of the quartz oscillator. The oscillation frequency of the QCM sensor decreases as the additional mass on the electrode surface increases; conversely, the oscillation frequency increases as the additional mass decreases. Therefore, a real-time monitoring function can be achieved by tracking the frequency shift in the QCM sensor, which can help to study the dynamic process of adsorption/desorption on the surface of the device. In addition, target gas molecules can be captured by depositing specific materials on the surface of the QCM device, thereby enabling the accurate measurement of various gas concentrations [11]. Therefore, QCM is an ideal device for gas detection at room temperature due to the advantages of simple construction, low cost, high sensitivity, easy preparation, and the low-frequency temperature coefficient [12].

In addition, the sensitive film of the gas sensors plays an important role in the gas adsorption/desorption process. In recent years, two-dimensional (2D) transition metal carbides/nitrides (MXenes) have been the subject of considerable interest due to their excellent electrical conductivity, tunable figure of merit, and abundant functional groups [13]. Ti_3_C_2_T_x_ is the most representative member of the MXene family, which has been widely used in energy storage [14], biosensing [15], photocatalysis [16], and gas sensing [17]. Nevertheless, the ease of oxidation of Ti_3_C_2_T_x_ MXene presents a challenge for MXene-based gas sensors, particularly in terms of long-term stability and selectivity.

On the other hand, zero-dimensional colloidal quantum dots (CQDs) show great potential for applications in photodetectors [18], gas sensing [19], and solar cells [20] due to their unique chemical and physical properties, extremely small size, low material cost, and adjustable energy band [21]. In particular, the large surface-to-volume ratio and the comparable crystal size to the Debye length provide a substantial number of active sites for gas adsorption, demonstrating the great potential of CQDs in gas sensing [22]. Recently, semiconductors of oxide and metal sulfide CQDs have been reported for the detection of NH_3_, NO_2_, and H_2_S gasses, including SnO_2_ [23], ZnO [24], PbS [25], and WO_3_ [26]. Among them, SnO_2_ has attracted much attention due to its good stability, suitable band gap, and abundant oxygen vacancies on the surface. Furthermore, due to the quantum effect, the surface effect becomes more significant when the size of SnO_2_ nanocrystals is reduced to a few nanometers to form quantum dots [27]. This facilitates the adsorption of gasses on the surface of the quantum dots [28]. However, gas sensors based on a single material frequently exhibit incompatibilities with regard to operating temperature, sensitivity, and detection limits. To resolve these issues, the overall performance of the sensor can be improved through combining low-dimensional nanomaterials and 2D MXene to form a heterojunction, such as Co_3_O_4_/Al_2_O_3_@Ti_3_C_2_Tx MXene [29], MXene/SnO_2_ [30], CuO/MXene [31], MXene/ZnO [32], and MXene/In_2_O_3_ [33]. The accumulation/depletion layer at the heterojunction interface provides favorable conditions for oxygen adsorption, resulting in an increased number of adsorbed oxygen molecules and an enhanced response of the gas sensor. Therefore, the modification of SnO_2_ CQDs on the Ti_3_C_2_T_x_ MXene surface is expected to achieve a high-performance ammonia sensor at room temperature.

In this paper, we prepared 0D SnO_2_ colloidal quantum dots and Ti_3_C_2_T_x_ MXene by solvothermal and in situ etching methods, respectively, and prepared the Ti_3_C_2_T_x_/SnO_2_ composites by mechanical stirring. The high-performance ammonia QCM sensor at room temperature was developed by depositing Ti_3_C_2_T_x_/SnO_2_ composites on the surface of the QCM device by spin-coating. This sensor achieves a highly accurate detection of sub-ppm NH_3_ at room temperature with high sensitivity and good selectivity. The QCM gas sensor can detect NH_3_ gas concentration in real time by monitoring the shift in the work frequency. Due to the synergistic effect of SnO_2_ and MXene, the gas response of Ti_3_C_2_T_x_/SnO_2_ is significantly improved compared to pure Ti_3_C_2_T_x_ MXene and SnO_2_ quantum dots.

## 2. Materials and Methods

### 2.1. Chemicals

All reagents were employed in their original unpurified state. The Ti_3_AlC_2_ MAX phase powder (99.99%, CAS: 196506-01-1) and LiF (99%, CAS: 7789-24-4) were procured from Jilin 11th Technology Co. Ltd. (Jilin, China) and Macklin Biochemical Technology Co., Ltd. (Shanghai, China). SnCl_4_·5H_2_O (99%, CAS: 10026-06-9), oleic acid (OA, CAS: 112-80-1), and oleyl amine (OLA, CAS: 112-90-3) were procured from Shanghai Aladdin Biochemical Technology Co., Ltd. (Shanghai, China). The HCl (CAS: 7647-01-0), toluene (CAS: 108-88-3), and ethanol (CAS: 64-17-5) were provided by Sinopharm Chemical Reagent Co., Ltd. (Shanghai, China). The QCM devices (frequency is 8 MHz) were bought from Shenzhen Jingyuanxing Electronics Co., Ltd., China, which consisted of an AT-cut quartz substrate and silver electrodes.

### 2.2. Preparation of Ti_3_C_2_T_x_ MXene

Pure accordion-like Ti_3_C_2_T_x_ MXene powder was successfully synthesized from the parent MAX phase powder (99.99%, Jilin 11 Technology, Changchun, China) using the in situ HF etching method, as reported in our previous work [34]. A total of 1.7 g LiF (99%, Macklin, Shanghai, China) was dissolved in 20 mL hydrochloric acid (9 M) by magnetic stirring. Then, 1 g Ti_3_AlC_2_ MAX phase powder was added to the above mixed solution and stirred in an oil bath of 40 °C for 24 h. The obtained mixture was centrifuged and washed repeatedly with deionized water until the pH value of the supernatant reached 7. Multilayered Ti_3_C_2_T_x_ MXene was dispersed in deionized water. Then, the suspension was dispersed in an ice bath by ultrasonic treatment for 2 h and centrifuged for 10 min. The Ti_3_C_2_T_x_ MXene powder was obtained by freeze-drying at −60 °C.

### 2.3. Synthesis of SnO_2_·CQDs and Ti_3_C_2_T_x_ MXene/SnO_2_ Composites

As shown in Figure 1, SnO_2_ colloidal quantum dots were synthesized by a conventional solvothermal method. A total of 0.596 g SnCl_4_·5H_2_O (99%, Aladdin, Jebel Ali, United Arab Emirates) was dissolved in 5 mL oleyl amine (OLA, Aladdin) and 20 mL oleic acid (OA, Aladdin) at 90 °C under a vacuum condition for 2 h. Then, 10 mL anhydrous ethanol was added to the above solution and well mixed. Afterwards, the mixed solution was transferred into a Teflon-lined autoclave to react at 180 °C for 90 min. Finally, the resulting product was centrifuged, washed three times with ethanol and toluene, and dispersed in toluene with a concentration of 20 mg/mL.

The Ti_3_C_2_T_x_ MXene was added to the SnO_2_ CQD solution and well mixed using magnetic stirring for 12 h at different weight ratios of 4 wt%, 8 wt%, 12 wt%, and 16 wt%. The solutions obtained are the Ti_3_C_2_T_x_/SnO_2_ composites at different weight ratios.

### 2.4. Characterization of Materials

Microstructure and surface morphologies of the Ti_3_C_2_T_x_ MXene, SnO_2_ CQDs, and Ti_3_C_2_T_x_/SnO_2_ composites were observed via field emission scanning electron microscopy (FE-SEM, Supra 55 Sapphire, Zeiss, Seoul, Republic of Korea) with an acceleration voltage of 15 kV. High-resolution transmission electron microscopy (HR-TEM, FEI Tecnai G2 F20) was used to investigate the crystalline structures of the materials. The crystal phases of the Ti_3_C_2_T_x_ MXene, SnO_2_ CQDs, and Ti_3_C_2_T_x_/SnO_2_ composites were analyzed using X-ray diffraction (XRD, MAXima XXRD-7000, Shimadzu, Kyoto, Japan) with a Cu Kα radiation source (40 kV and 40 mA) in the 2θ range of 3–60°. The chemical states of the sensitive films were studied via an X-ray photoelectron spectrometer (XPS, Escalab 250Xi, Thermo Fisher, Loughborough, UK).

### 2.5. Fabrication and Measurement of Ammonia QCM Sensors

The QCM devices (frequency was 8 MHz) were bought from Shenzhen Jingyuanxing Electronics Co., Ltd., Shenzhen, China, which consisted of an AT-cut quartz substrate and silver electrodes. The diameters of the electrodes and quartz substrate were 5 mm and 8 mm, respectively. In brief, the QCM devices were ultrasonically cleaned with acetone, anhydrous ethanol, and DI water, respectively. Then, 10 μL of SnO_2_/Ti_3_C_2_T_x_ MXene composites was spin-coated onto the QCM devices at 2000 rpm for 60 s, as shown in Figure 1. The sensors were named as pure SnO_2_, 4% Ti_3_C_2_T_x_/SnO_2_, 8% Ti_3_C_2_T_x_/SnO_2_, 12% Ti_3_C_2_T_x_/SnO_2_, and 16% Ti_3_C_2_T_x_/SnO_2_ for different weight amounts of added MXene (0 wt%, 4 wt%, 8 wt%, 12 wt%, and 16 wt%), respectively. The QCM resonant frequencies were measured before and after coating the sensitive layer, and the mass of material deposited on the electrodes was calculated using the Sauerbrey equation [35], as shown in Equation (S1). Appendix A provides a comprehensive illustration of the fundamental frequencies, frequency shifts, and load masses of all the sensors. As shown in Appendix A, following the deposition of the sensitive layer (SnO_2_, Ti_3_C_2_T_x_, Ti_3_C_2_T_x_/SnO_2_), a decrease in conductance was observed across all devices. Among them, SnO_2_-based QCM exhibited the lowest conductance, which can be attributed to the long carbon chains of OLA and OA capping on SnO_2_-inhibited carrier transport. The addition of Ti_3_C_2_T_x_ in Ti_3_C_2_T_x_/SnO_2_ composites markedly enhanced the conductance of the composites, facilitating the formation of a robust ohmic contact between the composites and the electrodes and ensuring the vibration of the QCM sensor. In addition, an increase in the conductance of the complex increases the Q value of the sensor, which in turn increases the range of detection, according to previous reports [36].

Frequency shifts in the sensors were measured in a chamber of 18 L in volume using QCM-I (Microvacuum Ltd., Budapest, Hungary) via a static sensing system (as illustrated in Figure 1). Dry NH_3_ gasses (a mixture of NH_3_ and N_2_) at concentrations of 2% were purchased from Gold Valley Gas Co., Ltd. (Shenzhen, China). A certain amount of 2% NH_3_ was injected into the 18 L chamber with an injector to obtain the corresponding concentration of the NH_3_ test environment. For each measurement, the sensor’s signal is acquired at a frequency of 1 Hz. The ammonia detection of the sensors was operated at room temperature (25 ± 1 °C) with a relative humidity (RH) of 60 ± 2%. The response time of the sensors is defined as the time to attain 90% of the maximum frequency shift upon the exposure of target gas. Similarly, the recovery time is defined as the time to return back to 10% of the frequency shift upon the exposure of air gas [37].

## 3. Results

### 3.1. Characterization of Sensitive Materials

As shown in Figure 2, the prepared Ti_3_C_2_T_x_ MXene exhibits a typical accordion-like multilayered structure, indicating that the Al layer in the Ti_3_AlC_2_ powder is completely removed. The inset of Figure 2a illustrates MXene with few layers prepared by ultrasonication. The typical layered two-dimensional structure of MXene facilitates rapid carrier transfer between the materials, leading to the rapid adsorption of gasses. The HR-TEM images (Figure 2b,c) display a lattice spacing of 0.25 nm, corresponding to the (006) planes of Ti_3_C_2_T_x_ MXene. The SEM image of pure SnO_2_ CQDs shown in Figure 2d shows uniform SnO_2_ sensitive films prepared by spin-coating, which ensures a good signal for the QCM device. The HR-TEM images of SnO_2_ quantum dots (Figure 2e,f) reveal uniformly sized and well-dispersed CQDs with a high degree of crystallinity. The lattice fringes with interplanar spacing values of 0.33 nm and 0.26 nm were observed, corresponding to the (110) and (101) planes of rutile SnO_2_, respectively.

The layered MXene can be clearly observed in the SEM image of Ti_3_C_2_T_x_ MXene/SnO_2_ composites (as shown in Figure 2g), indicating that Ti_3_C_2_T_x_/SnO_2_ composites were successfully prepared. Figure 2h illustrates the HR-TEM image of the Ti_3_C_2_T_x_/SnO_2_ composites, confirming that the SnO_2_ CQDs were anchored evenly over the layered Ti_3_C_2_T_x_ MXene surface without any aggregations. The selected area electron diffraction SAED pattern of Ti_3_C_2_T_x_/SnO_2_ composites (Figure 2i) reveals the diffractions of the (101), (110), and (211) crystal planes of SnO_2_, which further confirms the HR-TEM and the XRD results. Moreover, the EDS elemental mapping of Ti_3_C_2_T_x_/SnO_2_ composites (Figure 2j) shows an obvious and uniform distribution of Sn, O, Ti, and C elements, indicating the successful preparation of a Ti_3_C_2_T_x_/SnO_2_ hybrid heterointerface.

XRD patterns of the Ti_3_AlC_2_ MAX phase, Ti_3_C_2_T_x_ MXene nanosheets, pure SnO_2_ CQDs, and samples of Ti_3_C_2_T_x_/SnO_2_ composites are shown in Figure 3. Following the removal of the aluminum atomic layers from the Ti_3_AlC_2_ MAX powder, a shift in the (002) peak from 9.5° to 5.7° was observed, indicating an increase in the interlayer distance of Ti_3_C_2_T_x_ MXene nanosheets and successful preparation of Ti_3_C_2_T_x_ MXene [38]. The XRD patterns of Ti_3_C_2_T_x_ MXene that appeared at 5.7°, 24.7°, 38.7°, and 44.9° corresponded to the (002), (006), (008), and (106) planes, suggesting that the 2D Ti_3_C_2_T_x_ MXene nanosheets were successfully delaminated from Ti_3_AlC_2_ MAX.

The diffraction peaks of the pure SnO_2_ CQDs are completely consistent with the standard card of JCPDS no.41-1445, and no impurity peaks were observed, which indicates a good crystallinity and a high purity of SnO_2_ CQDs. The main peaks at 26.6°, 33.9°, and 51.8° correspond to (110), (101), and (211) planes of tetragonal rutile SnO_2_, respectively, which coincided with the HR-TEM results. As shown in the XRD patterns of all the Ti_3_C_2_T_x_/SnO_2_ composites, due to the low Ti_3_C_2_T_x_ MXene content proportion, the characteristic peaks of Ti_3_C_2_T_x_ are hardly observed in samples of 4% Ti_3_C_2_T_x_/SnO_2_ and 8% Ti_3_C_2_T_x_/SnO_2_. As the content proportion of Ti_3_C_2_T_x_ MXene in the composites increased from 12% to 16%, the intensity of the (008) and (106) peaks of Ti_3_C_2_T_x_ MXene, located at 38.7° and 44.9° in XRD patterns, respectively, increased gradually, indicating the successful preparation of the Ti_3_C_2_T_x_/SnO_2_ composites. The Raman spectra of the SnO_2_, Ti_3_C_2_T_x_, and 12% Ti_3_C_2_T_x_/SnO_2_ composites are shown in Appendix A. SnO_2_ exhibits three characteristic peaks at 317.1, 574.8, and 627.6 cm^−1^. The Ti_3_C_2_T_x_ MXene peaks are located at 408.1 and 606.8 cm^−1^ and correspond to the Eg vibrational modes of Ti with surface termination groups of -OH, -O, and -F [39]. The characteristic peaks of the Ti_3_C_2_T_x_/SnO_2_ composites correspond to the peaks of SnO_2_ and MXene, indicating that there were no vibrational changes during mixing and the composites were successfully prepared.

The surface valence bond states of the SnO_2_ CQDs and Ti_3_C_2_T_x_/SnO_2_ composites were analyzed by XPS, as shown in Figure 4. As shown in Figure 4a, the C 1 s spectra of the pure SnO_2_ CQDs show a C-C peak at 284.8 eV binding energies, which is attributed to adventitious C from the surrounding environment. The C 1 s energy spectrum of the Ti_3_C_2_T_x_/SnO_2_ composites presents four characteristic peaks of a C = O bond, C-O bond, C-C bond, and C-Ti, which are at 289.1 eV, 286.4 eV, 284.8 eV, and 281.4 eV, respectively [39]. The O 1s spectrum of SnO_2_ CQDs (Figure 4b) illustrates the deconvoluted peaks of binding energies at 533.8 eV, 532.5 eV, 531.9 eV, corresponding to chemisorbed oxygen (O_C_), oxygen vacancies (O_V_), and lattice oxygen (O_L_), respectively [40]. Compared to the SnO_2_ CQDs, the O 1s spectrum of the composites have an additional Ti-O deconvoluted peak located at 530.1 eV, indicating the successful composite of Ti_3_C_2_T_x_ Mxene and SnO_2_ CQDs. The O 1s results for the other Ti_3_C_2_T_x_/SnO_2_ samples are shown in Appendix A. Among them, the 12% Ti_3_C_2_T_x_/SnO_2_ had the highest O_V_ content of 58% (see Appendix A). The presence of oxygen vacancies (O_V_) can provide more sufficient active sites for gas adsorption, which is conducive to the adsorption of O_2_ molecules and target gasses. The Sn 3d spectrum (Figure 4c) shows a doublet of Sn 3d_2/3_ and Sn 3d_5/2_ at 495.8 eV and 487.3 eV. Figure 4d shows the Ti 2p spectrum, which is composed of Ti-C, Ti-O, Ti^2+^, and Ti^3+^. The deconvoluted peaks of binding energies at 454.7 eV/460.4 eV, 458.8 eV/464.5 eV, 455.7 eV/461.6 eV, and 458.0 eV/463.5 eV correspond to Ti-C, Ti-O, Ti^2+^, and Ti^3+^, respectively [41].

### 3.2. Sensing Performance of Sensors Based on Ti_3_C_2_T_x_/SnO_2_ Composites

The real-time responses of the sensors based on Ti_3_C_2_T_x_ MXene, SnO_2_ CQDs, and Ti_3_C_2_T_x_/SnO_2_ composites with different NH_3_ concentrations are shown in Figure 5a–c. When the sensors are exposed to ammonia, the work frequency of the sensors decreases, which is attributed to the mass loading effect [42]. Figure 5c illustrates the frequency response curves of the sensor of Ti_3_C_2_T_x_/SnO_2_ composites upon exposure to NH_3_ of different concentrations (twelve different NH_3_ concentrations ranging from 200 ppb to 50 ppm). The frequency shift in the sensor (12% Ti_3_C_2_T_x_/SnO_2_) to 50 ppm NH_3_ was −1140 Hz, which is 5.6 times larger than that of pure Ti_3_C_2_T_x_ MXene and 2.8 times higher than that of SnO_2_ CQDs. As the NH_3_ concentration decreased to 1 ppm, the sensor still exhibited a frequency shift in −120 Hz at room temperature, which is 7.9 times larger than the response of the MXene sensor. The specific response values at different NH_3_ concentrations are shown in Appendix A.

Figure 5d reveals the impact of composites with different Ti_3_C_2_T_x_ concentrations on the NH_3_ response of the sensor. As the Ti_3_C_2_T_x_ MXene content increases from 4% to 16%, the frequency response shows a trend of increasing and then decreasing. Compared to Ti_3_C_2_T_x_ and SnO_2_ sensors, the enhancement of the frequency response of the sensors of composites is attributed to the formation of a Ti_3_C_2_T_x_/SnO_2_ heterojunction. Among all the sensors based on Ti_3_C_2_T_x_/SnO_2_ composites, the 12% Ti_3_C_2_T_x_/SnO_2_ sensor exhibits the best performance. Therefore, the subsequent studies were conducted based on the sensor (12% Ti_3_C_2_T_x_/SnO_2_ composites).

Figure 6a illustrates a good liner relationship (R = 0.99) between the frequency shift and NH_3_ concentration (upon gas concentration in the range below 8 ppm) of the sensor-based 12% Ti_3_C_2_T_x_/SnO_2_ composites. The theoretical limit of detection (*LOD*) was calculated to be 73 ppb according to Equation (1).
(1)$$LOD\left(ppm\right)=3\frac{SD}{\left|s\right|}$$
where a standard deviation (SD) of 1.48 was obtained by averaging 300 data points from the baseline and s is the slope of the fitted curve in Figure 6a.

The response/recovery time of QCM sensors coated with Ti_3_C_2_T_x_, SnO_2_, and 12% Ti_3_C_2_T_x_/SnO_2_ exposed to 10 ppm NH_3_ was also evaluated (Figure 4b–d). The SnO_2_ sensor has a higher response than Ti_3_C_2_T_x_ sensors to 10 ppm NH_3_ mainly due to the large surface-to-volume ratio of SnO_2_ quantum dots and the large number of oxygen vacancies, resulting in a large amount of adsorbed oxygen involved in the gas-sensitive reaction. However, comparing both the SnO_2_-covered sensor and Ti_3_C_2_T_x_-covered sensor, Ti_3_C_2_T_x_ MXene showed a superior response/recovery time of 29/42 s. This is attributed to the high conductance of MXene facilitating the transfer of carriers in the sensitive film, promoting the rapid adsorption/desorption of target gas (Figure 6b,c). As illustrated in Figure 6d, the response of the Ti_3_C_2_T_x_/SnO_2_-based sensor is much higher than that of the two sensors above. In addition, the response/recovery time of the composite sensor was significantly shorter at 36/54 s compared to the SnO_2_-covered sensor due to the high conductance of MXene in the composites providing a channel for carrier transfer. The NH_3_ sensing performance of the QCM sensors in this work was summarized and compared with previously reported NH_3_ QCM sensors, as shown in Table 1. The results show that the sensor based on Ti_3_C_2_T_x_/SnO_2_ composites has significant advantages in terms of response value and response/recovery time.

Furthermore, the Ti_3_C_2_T_x_/SnO_2_ sensor was repeatedly exposed to 10 ppm NH_3_ at room temperature to evaluate the reproducibility. As shown in Figure 7a, the fluctuation in frequency responses is less than 6% during five cycles of detection for 10 ppm NH_3_, indicating the good reversibility and repeatability of the Ti_3_C_2_T_x_/SnO_2_ sensor. The selectivity of the Ti_3_C_2_T_x_/SnO_2_ sensors toward NH_3_ was evaluated by comparing other gasses (NO_2_, H_2_S, H_2_, ethanol, aniline, n-octylamine, and ethanolamine) at room temperature (Figure 7b). The frequency response of the sensor to other gasses could be negligible compared to the response to 50 ppm NH_3_, presenting excellent selectivity. Figure 7c illustrates the effect of environmental humidity on the response of the Ti_3_C_2_T_x_/SnO_2_ composite sensor exposure to 10 ppm NH_3_. The frequency response of the sensor decreases with the increase in the ambient relative humidity, which is mainly related to the adsorbed water molecules occupying the active sites of the Ti_3_C_2_T_x_/SnO_2_ composites in high humidity environments. The long-term stability of a gas sensor is also an important factor for its application. The long-term stability of the Ti_3_C_2_T_x_/SnO_2_ sensor was evaluated by comparing its response to 10 ppm NH_3_ at 60% RH for 30 days, as shown in Figure 7d. After 30 days, the frequency response of the sensor can still reach 91% of that of the initial sensor, indicating the potential applications of the Ti_3_C_2_T_x_/SnO_2_ sensor for NH_3_ sensing at room temperature. In addition, the long-term stability of the sensor was evaluated in high relative humidity (80% RH), as shown in Appendix A. Compared to pure Ti_3_C_2_T_x_, the Ti_3_C_2_T_x_/SnO_2_ composites showed more stability mainly due to the fact that the SnO_2_ covering the surface of the Ti_3_C_2_T_x_ blocked the oxygen gas direct contact with the Ti_3_C_2_T_x_ to reduce the oxidation of Ti_3_C_2_T_x_ MXene [17].

### 3.3. Sensing Mechanism of Sensor Based on Ti_3_C_2_T_x_/SnO_2_ Composites

It is well known that oxygen adsorption and the surface reaction on SnO_2_ are of fundamental importance in gas sensors. In air, the atmospheric oxygen molecules can trap free electrons from the conduction band of SnO_2_ and become ionized adsorbed oxygen ($O_{2\left(ads\right)}^{-}$ and $O_{\left(ads\right)}^{-}$) and form an electron depletion layer, resulting in band bending, as shown in Formulas (2)–(4) [51]. Figure 8 illustrates the NH_3_ sensing mechanism of the sensor based on Ti_3_C_2_T_x_/SnO_2_ composites. The improved performance of the composites-based sensors can be attributed to the Schottky barriers created in the Ti_3_C_2_T_x_/SnO_2_ composites. Figure 8a displays the band diagram of SnO_2_ and Ti_3_C_2_T_x_ before the contact. Ti_3_C_2_T_x_ MXene has excellent conductivity and metallic properties with a work function of 3.9 eV [52]. SnO_2_ is a typical n-type semiconductor with a work function of 4.9 eV [53]. After contact, since the work function of Ti_3_C_2_T_x_ is lower than that of SnO_2_, electrons flow from Ti_3_C_2_T_x_ to SnO_2_, bending the energy bands until reaching Fermi-level equilibrium (Figure 8b). In addition, an electron accumulation layer was created at the heterojunction interface [6]. This causes oxygen molecules to capture more free electrons from the Ti_3_C_2_T_x_/SnO_2_ heterojunction, forcing more O_2_ to be adsorbed onto the Ti_3_C_2_T_x_/SnO_2_ surface. This results in a higher density of oxygen adsorbed on the surface of the nanofilm and therefore a more intense depletion of the charge carriers from its lattice (see Figure 8c). When the Ti_3_C_2_T_x_/SnO_2_ sensor is exposed to NH_3_, these additional $O_{2\left(ads\right)}^{-}$ and $O_{\left(ads\right)}^{-}$ participate in the NH_3_ sensing reaction, enhancing the performance of the sensor. Upon exposure to target reduced NH_3_, the adsorbed oxygen ions ($O_{2\left(ads\right)}^{-}$ and $O_{\left(ads\right)}^{-}$) react with the NH_3_ molecules, as shown in Equations (5) and (6) [6], releasing electrons back into the SnO_2_ (Figure 8d).
(2)$$O_{2\left(gas\right)}\to O_{2\left(ads\right)}$$
(3)$$O_{2\left(ads\right)}+e^{-}\to O_{2\left(ads\right)}^{-}$$
(4)$$O_{2(ads)}^{-}+e^{-}\to O_{\left(ads\right)}^{-}$$
(5)$${4NH}_{3}+{3O}_{2(ads)}^{-}\to {2N}_{2}+{6H}_{2}O+{3e}^{-}$$
(6)$${2NH}_{3}+{3O}_{\left(ads\right)}^{-}\to N_{2}+{3H}_{2}O+{3e}^{-}$$

In addition, the presence of oxygen vacancies (O_V_) can provide more sufficient active sites for gas adsorption, which is conducive to the adsorption of O_2_ molecules and target gasses, thus promoting the adsorption behaviors of the sensor [53]. From the XPS analysis results (Figure 4b and Appendix A), it can be seen that the content of O_V_ on the surface of the Ti_3_C_2_T_x_/SnO_2_ composites firstly increases and then decreases with the increase in the content of Ti_3_C_2_T_x_, and the Ti_3_C_2_T_x_/SnO_2_ has the highest content of oxygen vacancies (see Appendix A). The presence of a large number of O_V_ on the surface of the composites can further enhance the NH_3_ sensing performance of the Ti_3_C_2_T_x_/SnO_2_ sensor, which explains why the 12% complex has the best sensor performance. This also confirm the sensing mechanism for the enhancement of the Ti_3_C_2_T_x_/SnO_2_ composites.

## 4. Conclusions

In summary, we have prepared a high-performance ammonia QCM sensor based on Ti_3_C_2_T_x_/SnO_2_ composites synthesized by a solvothermal method. By evaluating the effect of the Ti_3_C_2_T_x_ content proportion (in the range of 4~16 wt%) on the performance of the sensor, an optimal Ti_3_C_2_T_x_ content of 12 wt% was determined. Upon exposure to 50 ppm NH_3_, the sensor based on 12 wt% Ti_3_C_2_T_x_/SnO_2_ composites exhibited an excellent frequency shift (Δf = −1140 Hz) at room temperature. In addition, the sensor shows a good linearity (R = 0.99) at low NH_3_ concentrations (from 200 pbb to 8 ppm), with a theoretical detection limit of 73ppb. The sensor has good long-term stability and repeatability. The excellent performance of this sensor is attributed to the establishment of a Ti_3_C_2_T_x_/SnO_2_ heterojunction, which provides ideas to further promote the application of composite materials in gas sensors. Although there exists a superiority of composites as sensing materials for QCM gas sensors, large-scale manufacturing remains as a challenge before the practical application of nanomaterial-based QCM sensors. Additionally, the miniaturization of the sensor also aligns with the prevailing trend in gas sensor design. Consequently, there is considerable scope for further innovation in composite-based QCM gas sensors, particularly in enhancing selectivity, optimizing reproducibility, and developing miniaturized fabrication techniques.

## Figures and Tables

**Figure 1 nanomaterials-14-01835-f001:**
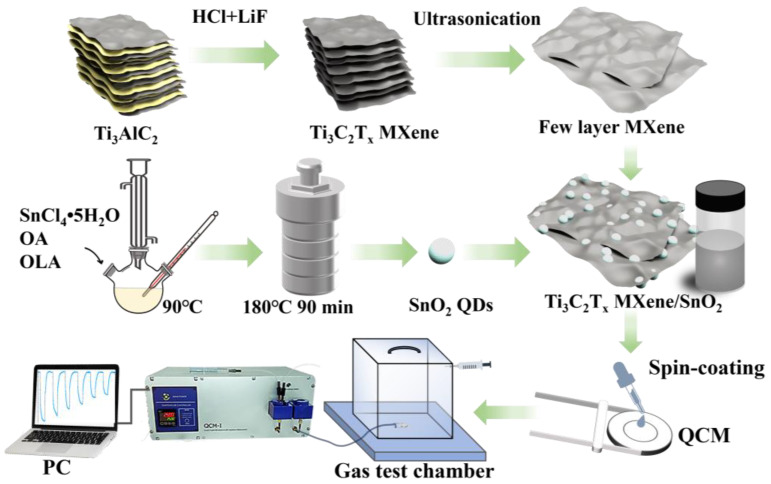
Schematics of the synthesis of Ti_3_C_2_T_x_/SnO_2_ composites and preparation of ammonia QCM sensors.

**Figure 2 nanomaterials-14-01835-f002:**
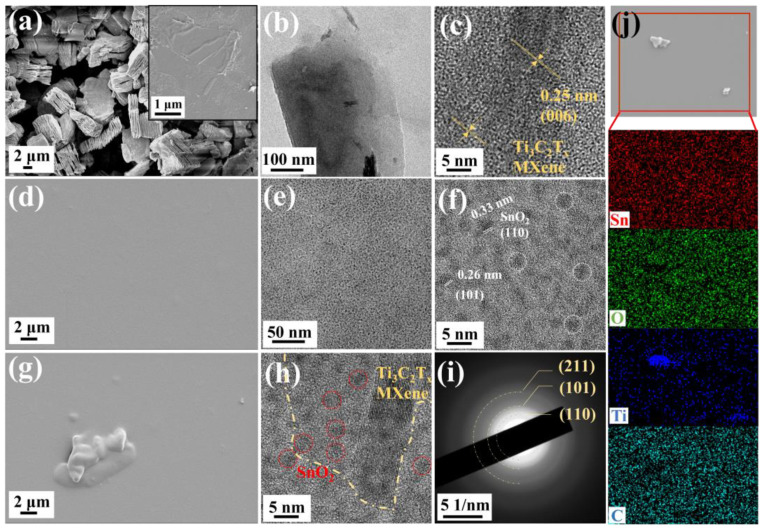
SEM images of (**a**) accordion-like Ti_3_C_2_T_x_ MXene, (**d**) SnO_2_ CQDs, and (**g**) Ti_3_C_2_T_x_/SnO_2_ composites. Inset of (**a**) is SEM image of MXene with few layers. HR-TEM images of (**b**,**c**) MXene, (**e**,**f**) SnO_2_ CQDs, and (**h**) Ti_3_C_2_T_x_/SnO_2_ composites. (**i**) SAED pattern of Ti_3_C_2_T_x_/SnO_2_ composites. (**j**) EDS map scanning analysis of Sn, O, Ti, and C elements.

**Figure 3 nanomaterials-14-01835-f003:**
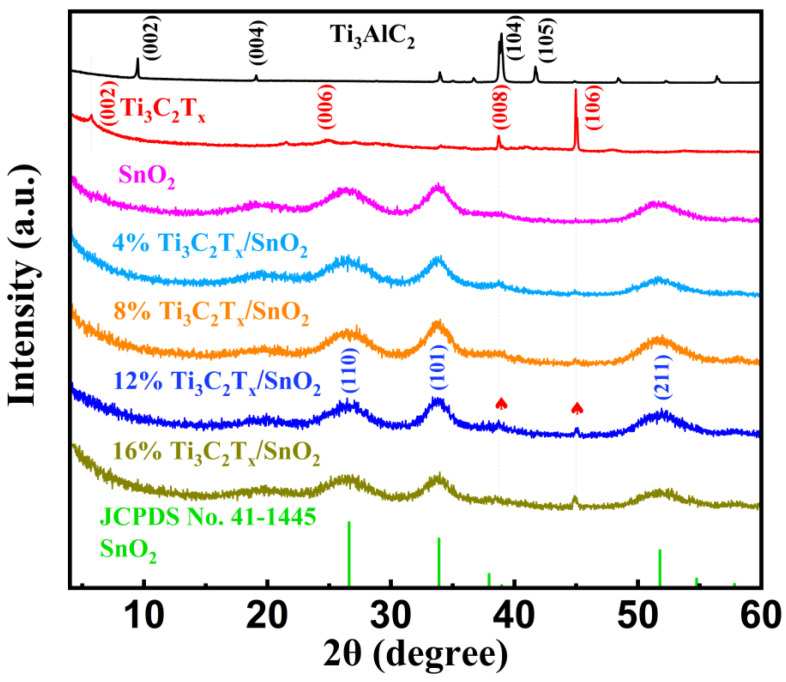
XRD patterns of the Ti_3_AlC_2_ MAX phase, accordion-like Ti_3_C_2_T_x_ MXene, pure SnO_2_ CQDs, and Ti_3_C_2_T_x_/SnO_2_ composites.

**Figure 4 nanomaterials-14-01835-f004:**
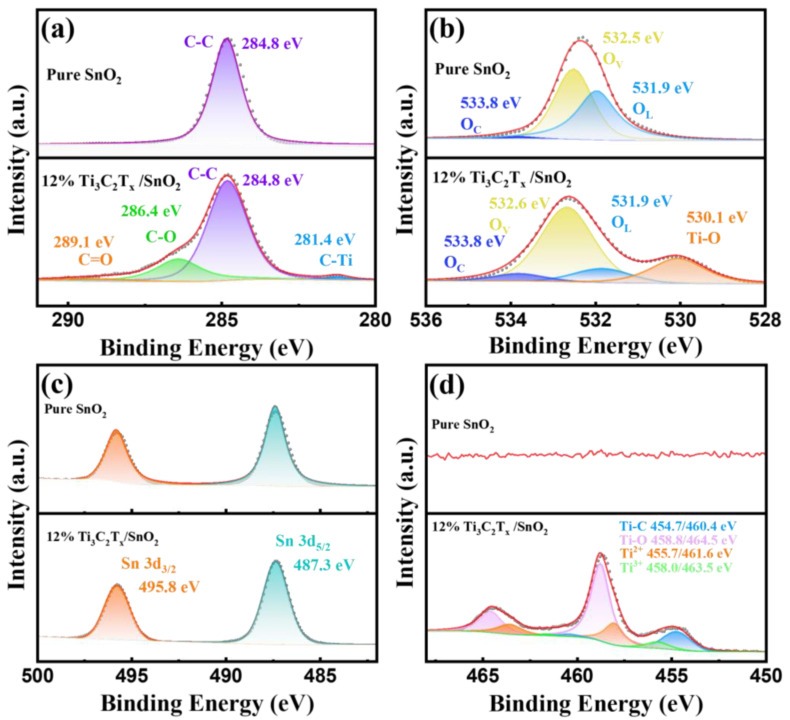
XPS spectra of (**a**) C 1s, (**b**) O 1s, (**c**) Sn 3d, and (**d**) Ti 2p.

**Figure 5 nanomaterials-14-01835-f005:**
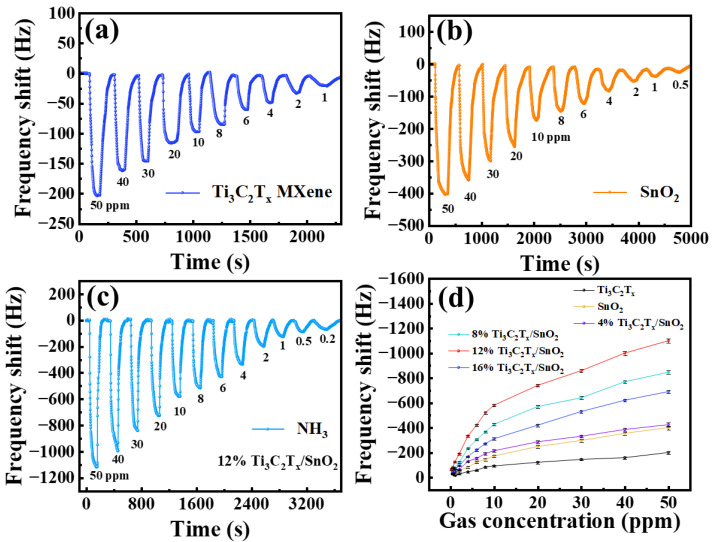
Response-time curves with different NH_3_ concentrations based on (**a**) a Ti_3_C_2_T_x_ sensor, (**b**) a SnO_2_ sensor, (**c**) a Ti_3_C_2_T_x_/SnO_2_ sensor. (**d**) The curves between frequency shift and gas concentration of all sensors.

**Figure 6 nanomaterials-14-01835-f006:**
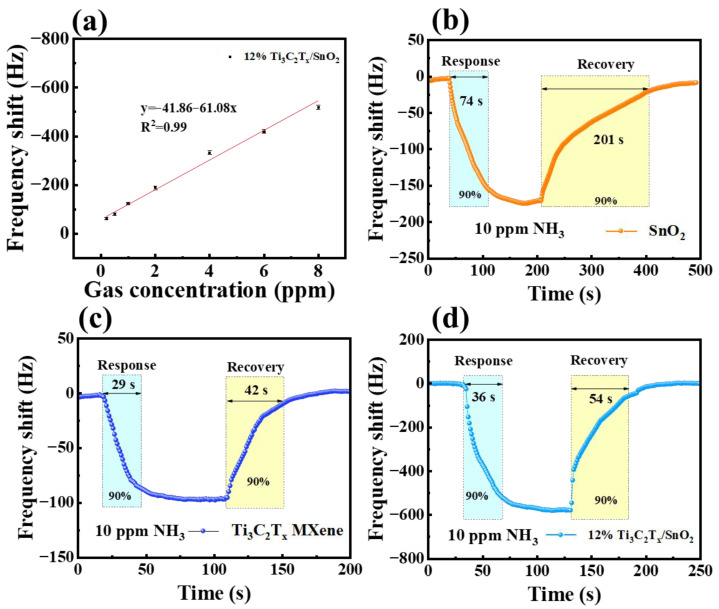
(**a**) Frequency shift of Ti_3_C_2_T_x_/SnO_2_ sensor to NH_3_ concentrations varying from 200 ppb to 8 ppm. Response and recovery curves of (**b**) SnO_2_, (**c**) Ti_3_C_2_T_x_, and (**d**) Ti_3_C_2_T_x_/SnO_2_ composites.

**Figure 7 nanomaterials-14-01835-f007:**
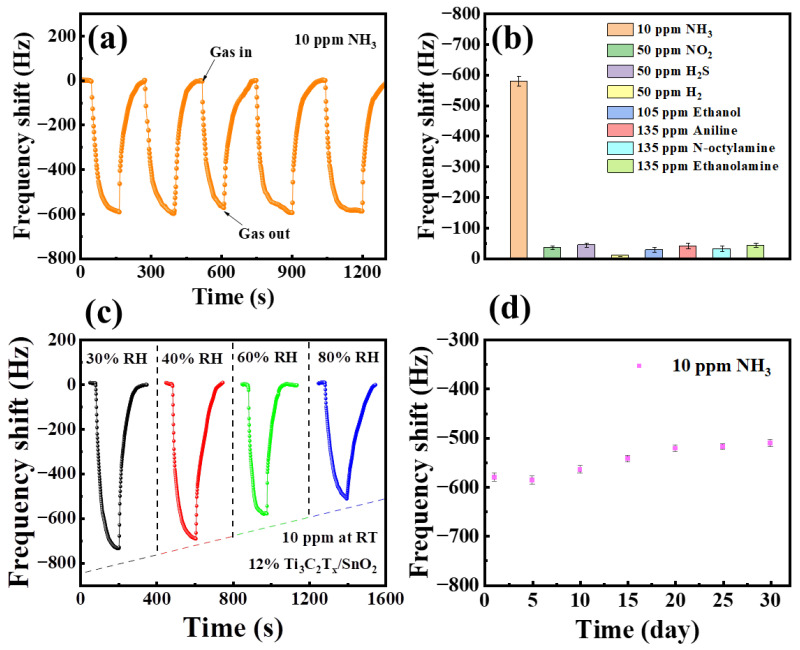
(**a**) Repeated response of Ti_3_C_2_T_x_/SnO_2_ sensor to 10 ppm NH_3_. (**b**) Selectivity of sensor based on Ti_3_C_2_T_x_/SnO_2_ composites. (**c**) Response curves of Ti_3_C_2_T_x_/SnO_2_ sensor to 10 ppm of NH_3_ at different relative humidities. (**d**) Long-term stability of sensor based on Ti_3_C_2_T_x_/SnO_2_ composites.

**Figure 8 nanomaterials-14-01835-f008:**
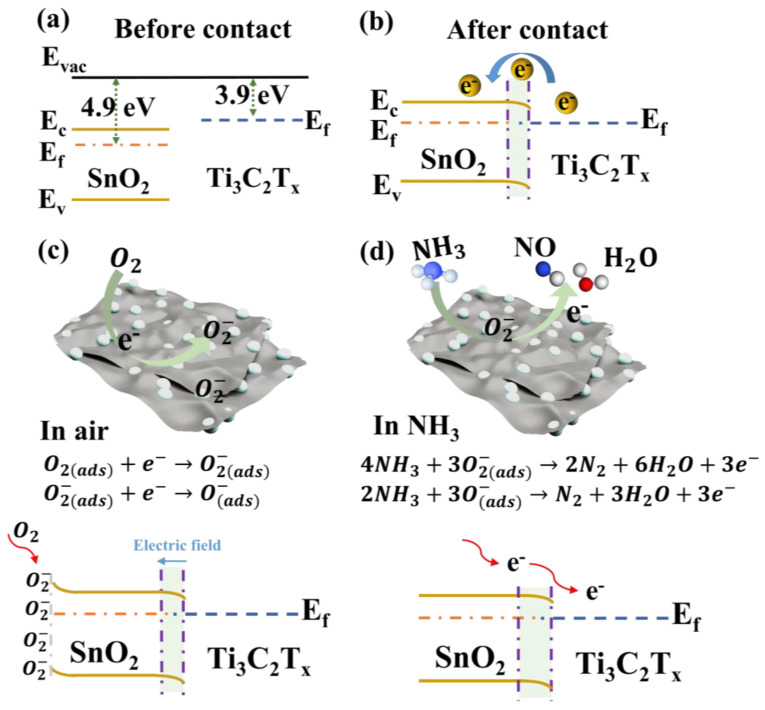
Energy band diagrams of SnO_2_ and Ti_3_C_2_T_x_ (**a**) before and (**b**) after contact. Schematic diagram of NH_3_ sensing mechanism of sensor based on Ti_3_C_2_T_x_/SnO_2_ composites, (**c**) in air, and (**d**) exposure to NH_3_.

**Table 1 nanomaterials-14-01835-t001:** Performance comparison of room temperature NH_3_ QCM sensors based on different composites.

Material	Gas Concentration (ppm)	Frequency Shift (Hz)	Response/Recovery Time (s)	Reference
V_2_O_5_ nanoplatelets	9	−45	2/28	[43]
ZnO nanorods	200	−24	-/1800	[44]
ZnO/MXene	100	−131	-/40	[45]
PAH/PAA	200	−120	50/420	[36]
PVP/GO	100	−90	>3 min	[46]
Boric acid-doped PVAc	100	72	-	[47]
Ti_3_C_2_T_x_-S	30	−600	33/58	[39]
PVAc/CA	50	−147	31/200	[48]
PSS/ZIF-C/PANI	100	−567	28/17	[49]
Carbon nanocoil	200	34	-/119	[50]
Ti_3_C_2_T_x_/SnO_2_	10	−596	36/54	This work

## Data Availability

All data, models, and codes generated or used during the study appear in the submitted article.

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
