# Peer review of "High-Performance Ammonia QCM Sensor Based on SnO2 Quantum Dots/Ti3C2Tx MXene Composites at Room Temperature"

_nanomaterials, 2024, doi:10.3390/nano14221835_

Round 1
Reviewer 1 Report
Comments and Suggestions for Authors
This paper report the study and the development of a QCM sensor to measure NH3 where a Ti3C2Tx/SnO2 composite material works as interaction membrane. The authors used a surface modification of SnO2 colloidal quantum dots to develop the coating. They present material preparation and sensor tests. In addition they report a table aimed to comparison of the proposed sensor and other in literature.
In my opinion the manuscript report more accurately the research and the methods. As following reported, in my opinion, the manuscript needs some major revisions before the publication:
-In section 2.4 , where the authors described QCM they reports that the electrodes were made of silver. The authors should add some comments about the possibility that silver oxide was on the surface and the it possible role during the coating process.
- The authors should add comments about the possible role that paly the electrode material during the developed material deposition and the possible chemical interaction with this type of electrodes.
- Is it possible to use different electrodes materials ? please add some comments.
- The authors should add comments and details about the method that they have used to test the developed sensor. In particular NH3 cylinder was used ? in air or in nitrogen ….. or other method has been used to generate the ammonia ?
- the authors should add detail about the resolution of frequency acquisition system to measure the frequency shift of QCM.
- The authors affirmed that the sensor worked at room temperature (25 °C). But some comments about the influence of “detection/interaction process” behaviours vs temperature changing should be added.
- The authors should add some comments about the interaction of the presented material with “ammine family” not only NH3 for example: trimethylamine, aniline, …
Reviewer 2 Report
Comments and Suggestions for Authors
The authors present SnO2 and Mxene composite NH3 gas sensors. The research idea is well-conceived, and the manuscript is well-written. The results are also novel and supported by the conclusions and references. It would be an excellent contribution to Nanomaterials. Hence, I would like to accept after the minor revision of the following comments:
1- Define the abbreviations at the first instance of their usage in the manuscript.
2- The novelty of the research idea should be indicated in the abstract of the introduction section.
3- Provide the CAS numbers of all the chemicals, solutions and substrates used in the experiments.
4- Did the authors collect the powder or deposit the materials on a substrate?
5- Provide the JCPDS reference for XRD.
6- Provide the limitations of this study and the future directions in the conclusion section.
Reviewer 3 Report
Comments and Suggestions for Authors
1. While the sensing mechanism is discussed, it lacks a deeper theoretical exploration of the interaction between NH3 molecules and the SnO2/Ti3C2Tx composite, particularly at the molecular level. A more detailed analysis of the heterojunction formation and the electronic interaction could be beneficial.
2. The paper mentions different weight ratios for Ti3C2Tx/SnO2 but does not provide sufficient information on how the optimal ratio was chosen. A discussion of why 12 wt% was the best-performing ratio would enhance the reader's understanding.
3. Although XRD and XPS are utilized, other techniques like Raman spectroscopy could provide additional insights into the structural properties and chemical states of the composite materials. Incorporate a comparative analysis of the performance of this sensor with other recent materials not only in terms of sensitivity but also considering selectivity and long-term durability.
4. The stability of the sensor over 30 days is briefly mentioned. However, more in-depth data on the potential degradation mechanisms or stability in various environments (e.g., humidity changes) would be beneficial.
5. Expand on the electron transfer process at the heterojunction interface. Diagrams or simulations showing the band alignment and electron movement could help visualize the sensing mechanism.
6. Provide a more detailed statistical analysis of the experimental data, including error margins, standard deviations, or confidence intervals where appropriate.
7. The number of reference papers used for comparison in Table 1 should be further increased, with at least 10 or more.
8. The introduction section's description of QCM is too simplistic.
9. The content regarding the gas sensing mechanism is too simplistic.
Round 2
Reviewer 1 Report
Comments and Suggestions for Authors
The authors have added some comments to answer to my questions. In addition they have report additional figure and have changed some paper sections to enhance the study and test comprehension. In my opinion the paper now is ready to be published
Author Response
Comment:The authors have added some comments to answer to my questions. In addition they have report additional figure and have changed some paper sections to enhance the study and test comprehension. In my opinion the paper now is ready to be published.
Rseponse: We would like to thank the Reviewers for their recognition of the manuscript. Thank you very much for your help and precious time.
Reviewer 3 Report
Comments and Suggestions for Authors
The author has carefully revised the manuscript according to the suggestions, and it is now ready for acceptance.
Author Response
Comment: The author has carefully revised the manuscript according to the suggestions, and it is now ready for acceptance.
Rseponse: We would like to thank the Reviewers for their recognition of the manuscript. Thank you very much for your help and precious time.